# Fatty acid metabolism changes in association with neurobehavioral deficits in animal models of fetal alcohol spectrum disorders

Hye Mee Hwang[1,2], Yuka Imamura Kawasawa[3,4], Aiesha Basha[1], Shahid Mohammad[1], Mariko Ito[1] & Kazue Hashimoto-Torii [1,5 ✉]

Fetal alcohol spectrum disorders (FASD) show behavioral problems due to prenatal alcohol exposure (PAE). A previous study reports changes in gene expressions linked to fatty acid (FA) metabolism in the cerebral cortex of the PAE mouse model. We find an increase of palmitic acid and arachidonic acid in phospholipid in the cerebral cortex of PAE at postnatal day 30. The increase of palmitic acid is consistent with increase of the producing enzyme, *Fasn* (fatty acid synthase). Decrease of 26:6 FA is also consistent with the increase of the enzyme which uses 26:6 as a substrate for making very long chain FAs, *Elovl4* (elongation of very long chain fatty acids protein 4). However, there is no increase in the elongated products. Rather, lipid droplets (LDs) accumulated in the brain. Although FA-associated metabolic measurements are not affected by PAE, the abundance of FA-related gut microbiota is altered. This suggests that the gut microbiome could serve as a tool to facilitate uncovering the brain pathophysiology of FASD and a potential target to mitigate neurobehavioral problems.

[1] Center for Neuroscience Research, The Children's Research Institute, Children's National Hospital, Washington, DC, USA. [2] The Institute for Biomedical Sciences, School of Medicine and Health Sciences, The George Washington University, Washington, DC, USA. [3] Department of Pharmacology, Pennsylvania State University College of Medicine, Hershey, PA, USA. [4] Department of Biochemistry and Molecular Biology, Institute for Personalized Medicine, Pennsylvania State University College of Medicine, Hershey, PA, USA. [5] Departments of Pediatrics, and Pharmacology & Physiology, School of Medicine and Health Sciences, The George Washington University, Washington, DC, USA. ✉email: khtorii@childrensnational.org

Alcohol exposure during gestation adversely affects fetal development. Fetal alcohol spectrum disorder (FASD) is an umbrella term describing a group of clinical conditions resulting from prenatal alcohol exposure (PAE). Those conditions include facial dysmorphology and cognitive and neurobehavioral deficits[1–5]. Recent clinical and preclinical animal studies demonstrated that PAE is associated with metabolic dysregulation after birth, including increased body fat, a higher incidence of obesity, and hypertension[6–9], highlighting the metabolically vulnerable individuals with PAE. Of note, metabolic disruption was shown to associate negatively with cognition in adolescents[10] and positively with the risk of developing dementia such as Alzheimer's disease (AD) in adults[11]. Disrupted lipid homeostasis was reported in the brain biopsy of other neurodegenerative diseases, including amyotrophic lateral sclerosis and Parkinson's diseases (PD)[12–14]. In these patients, accumulation of lipid droplets (LDs) was found in various brain regions and cell types[13–15]. LDs are organelles that store excess fatty acids to protect cells from lipid toxicity[16], and the biogenesis and degradation of LDs are tightly coupled with cellular metabolism to maintain homeostatic lipid levels[17].

Alcohol interacts directly with fatty acid to produce fatty acid ethyl ester, thereby identified as a biomarker of maternal alcohol drinking[18]. Furthermore, in pregnant mothers who drink alcohol, fatty acid composition in their plasma was different between mothers with offspring showing abnormal development and those showing normal development[19]. In animals exposed to alcohol throughout gestation, docosahexaenoic acid (DHA), an n-3 fatty acid, was reduced in phospholipid collected from the postnatal hippocampus[20,21]. In another study, administration of DHA between P11 and P20 improved social behavior deficits of PAE rats[22]. In FASD young children, choline that enhances the fatty acid oxidation in the liver[23] was supplied for 9 months, and improved non-verbal IQ and working memory in those children[24]. These studies collectively support targeting disturbed fatty acid metabolism for potential to mitigate lipid dyshomeostasis in FASD. Gut microbiota crucially regulates host metabolism[25] and modulates brain function via the gut-brain axis[26]. Short chain fatty acids (SCFAs) produced by gut microbiota are one of the major metabolites that influence host lipid metabolism, and accumulating evidence demonstrates that the SCFA also affects neurobehavior through the gut-brain axis[27,28] by regulating the biogenesis, oxidation, and metabolisms of fatty acids in various tissues, likely in context dependent manner[29]. Some types of SCFAs inhibit the functions of a histone deacetylase to improve memory and learning in normal animals[30,31] and an animal model of meningitis[32], while others lead to autism-like behavior issues[33,34]. A number of studies also found differential abundances of gut microbial compositions in the feces of patients who suffer from neurodegenerative diseases[35]. Animal studies also demonstrated that changing gut microbial composition affects neuropathology and behavior[36,37].

Similar to the adult cases described above, several studies have reported that both maternal and early postnatal microbiota are essential for exploratory, social, and sensorimotor behavior development in mice[38,39]. Depleting maternal microbiota by antibiotic treatment impaired thalamocortical axonogenesis of the fetus, and the replenishment of the Clostridia bacteria into the mother by oral gavage improved the impairment[40]. In FASD, there are a few recent studies reporting changes in the gut microbiome in animal models of FASD[41–43]. However, how these changes are associated with the neurobehavioral phenotypes remains unknown.

Using a mouse model of PAE, this study revealed that learning deficits and anxiety phenotype are associated with the disturbance of fatty acid metabolism in the brain and gut microbiota, but not with other metabolic measurements such as body weight, plasma triglycerides, or blood glucose levels. These results suggest that the gut microbiome may serve as a sensitive biomarker for lipid dysregulation and have crucial contributions to lipid-mediated brain pathology and neurobehavioral issues, irrespective of blood metabolic molecules in FASD.

## Results

### PAE increases the expression of a fatty acid synthesizing enzyme in the frontal cortex at a juvenile stage.
In our previous study, single cell RNA sequencing revealed that the expressions of genes in the fatty acid biosynthesis pathway, such as elongation of very long chain fatty acids protein 4 (Elovl4) and fatty acid synthase (Fasn), were upregulated in the cortical neurons in the mouse primary motor cortex at P30, long after acute prenatal alcohol exposure once a day at 4.0 g/kg weight at embryonic day 16 and 17. The offspring, a model of FASD, does not have obvious morphological abnormality or changes in the density of neurons in the brain[44]. ELOVL4 is a fatty acid elongase required to form long chain fatty acids greater than 28 carbon lengths long[45]. To examine whether the protein level is also increased, the number of ELOVL4-positive cells was compared between PAE and the control brains (offspring of PBS received dams) by immunohistochemistry at P30 when both motor skill learning deficits and anxiety were observed in PAE mice[44,46].

Similar to the increase at the RNA level[44], the number of cells that express ELOVL4 proteins was increased in the PAE primary motor cortex (Fig. 1a, b). We also found that a nearby cortical region, the cingulate cortex that is involved in controlling anxiety[46–48], shows an increase in the number of ELOVL4-positive cells in PAE mice (Fig. 1d), suggesting the increase of ELOVL4 in the frontal cortex of PAE mice. However, the number of ELOVL4 expressing cells is not altered in the piriform cortex, striatum, lateral ventricular wall and hippocampus (Supplementary Fig. 1). Consistent with the previous mouse study in which ELOVL4 protein expression was primarily expressed in neurons with little to no expression in astrocytes in normal adult mouse brains[49], most of the ELOVL4 immunolabeling was found in the soma of cells that express NeuN, a marker of mature neurons, in control animals (Fig. 1c). In addition, widespread distribution of ELOVL4 in the cytoplasm is similar between PAE and the control (Fig. 1b, e, g). Given that PAE did not alter the number of ELOVL4 expressing cells in the cerebellum (Fig. 1f) that also controls motor learning[50], fatty acid composition and/or metabolism are anticipated to be altered uniquely in neurons of the frontal cortex of PAE mice.

### The fatty acid composition of phosphatidylethanolamine in the cell membrane is altered in PAE motor cortex at the juvenile stage.
We then examined the fatty acid compositions in the cell membrane in the motor cortex at P30 by analyzing fatty acid in phospholipid using liquid chromatography tandem mass spectrometry. Phosphatidylcholine (PC) and phosphatidylethanolamine (PE) are the most abundant phospholipids in cell membranes[51]. PC is found mostly in the outer layer of the plasma membrane, whereas PE is found in the inner layer of the plasma membrane and mitochondrial membrane[51]. Therefore, we analyzed the fatty acids in both PC and PE.

First, we examined total amounts of PC and PE and found no changes in PAE mice compared to control mice (Supplementary Fig. 2a, b). Disturbance of the PC/PE ratio is known to alter energy metabolism and is associated with non-alcoholic fatty liver diseases and obesity[51]. Therefore, we examined the PC/PE ratio. Similar to another study[52], the PC/PE ratio indicated a higher amount of PE than PC in cortices of both control and PAE groups. However, their ratio was not altered by PAE (Supplementary Fig. 2c).

In the comparisons of each fatty acid molecular species, a increase was found only in the palmitic acid (16:0) and the arachidonic acid (20:4) in PE of PAE mice (Fig. 2a). Palmitic and

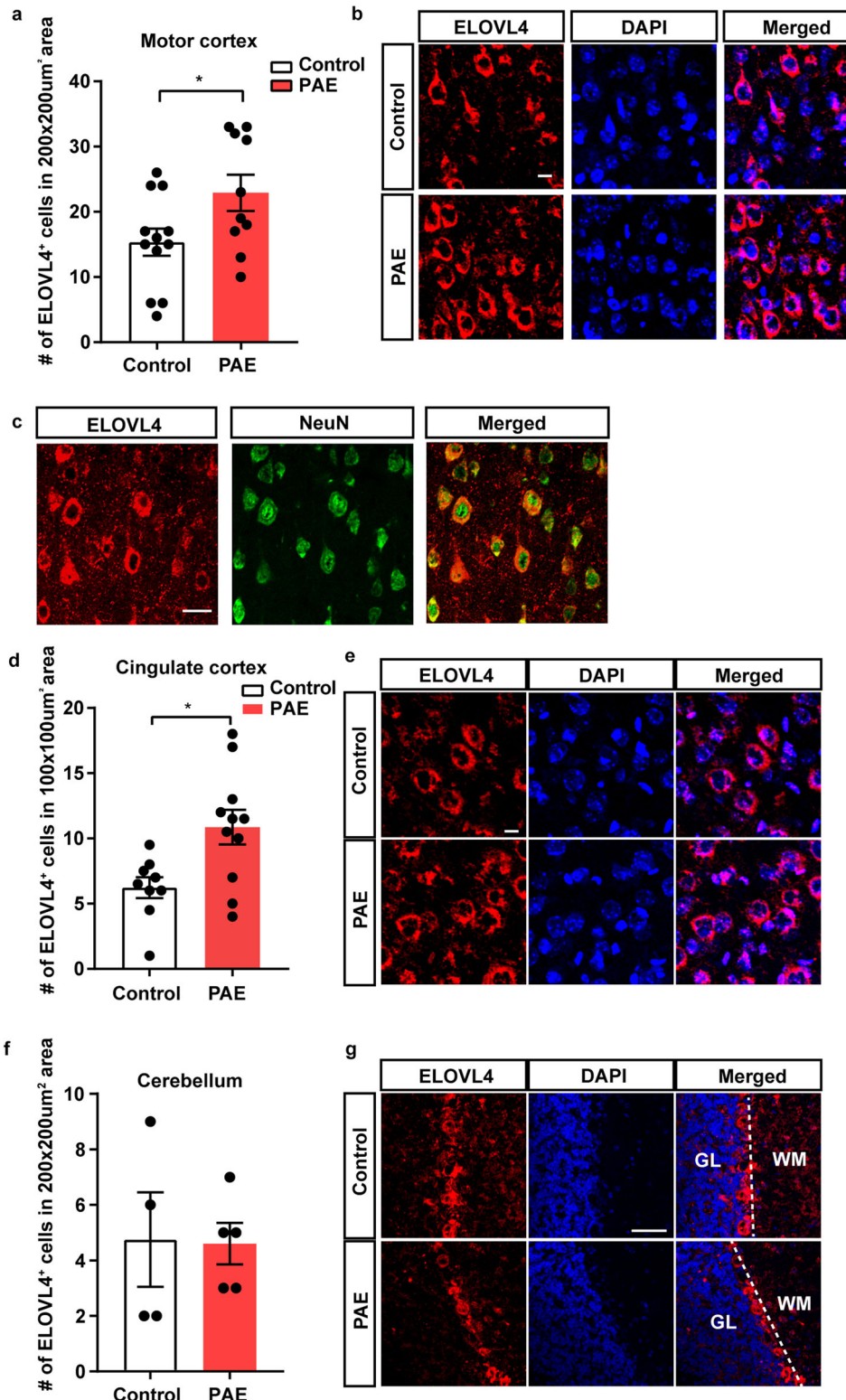

**Fig. 1 PAE increases neuronal ELOVL4 expression in the motor and cingulate cortices at P30. a** The number of ELOVL4-positive cells is higher in the motor cortex of PAE mice compared to the control mice ($p = 0.038$). $^*p < 0.05$. Student's $t$-test. Control $n = 12$, PAE $n = 10$. **b** Representative images for immunolabeling of ELOVL4 (red) and DAPI (blue). Scale bar $= 10 \, \mu m$. **c** Representative images of coronal section from control cortex demonstrate that ELOVL4 expression (red) is detected in NeuN-positive neurons (green). Scale bar $= 25 \, \mu m$. **d** The number of ELOVL4-positive cells is higher in the cingulate cortex of PAE mice compared with control mice at P30 ($p = 0.011$). $^*p < 0.05$. Student's $t$-test. Control $n = 9$, PAE $n = 11$. **e** Representative images of ELOVL4 (red) and DAPI (blue) staining in the cingulate cortex. Scale bar $= 10 \, \mu m$. **f** The number of ELOVL4-positive cells in the cerebellum is not different between control and PAE mice at P30. Control $n = 4$, PAE $n = 5$. **g** Representative images of ELOVL4 immunostaining in the cerebellum. Scale bar $= 50 \, \mu m$. Graphs represent mean ± SEM. Each dot represents an individual animal. GL granular layer, WM white matter.

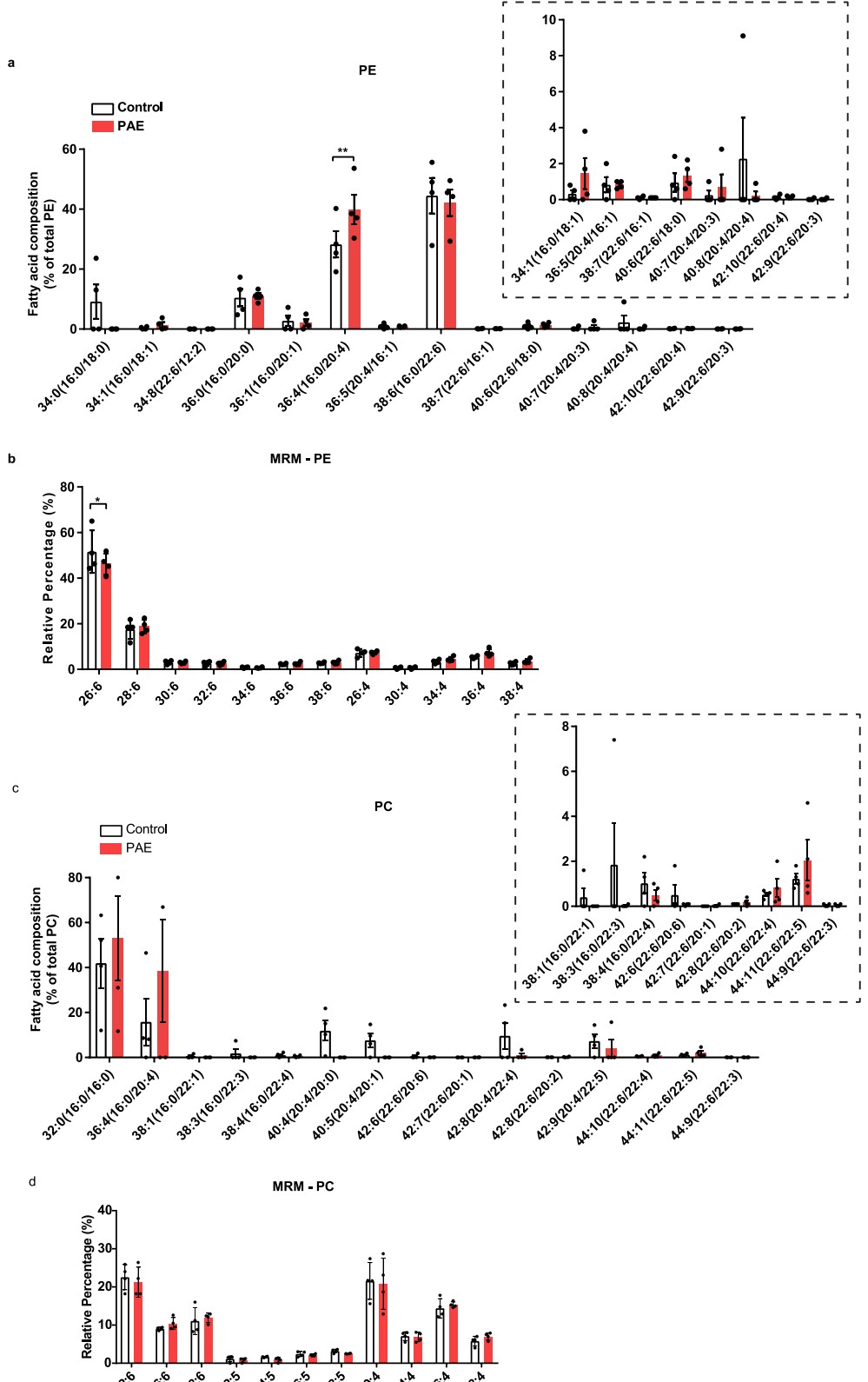

arachidonic acids are saturated fatty acids (SFA) and polyunsaturated fatty acids (PUFA), respectively, both of which are highly abundant in the cell membrane[53,54]. The increase in palmitic acid was consistent with an increase in *Fasn* RNA expression[44] as the function of the FASN is to synthesize palmitic acid[55]. In PC, although the palmitic acid and the arachidonic acid show increase trend similar to that of PE in PAE, there was no difference in any

of the fatty acid molecular species between PAE and control mice (Fig. 2c).

The Multi reaction monitoring (MRM) was used to detect very long chain fatty acids that ELOVL4 specifically produces in both PC and PE. ELOVL4 is involved in elongating both SFA and PUFA by adding 2 additional carbons to synthesize very long chain fatty acids that are longer than 28 carbons (C28); thus, any

**Fig. 2 Fatty acid composition in PE, but not in PC, is altered by PAE in the motor cortex. a** PAE mice show an increase in abundance of 36:4 of PE (16:0/20:4) compared to control mice ($p = 0.0096$) in the motor cortex at P30. Inset shows the magnified view of the fatty acid species that are less abundant. **b** Multi reaction monitoring (MRM) analysis reveals a decrease of 26:6 of PE fatty acid in PAE motor cortex compared to control ($p = 0.0397$). No interaction between fatty acid species and prenatal exposure types by two-way ANOVA. Bonferroni's multiple comparisons tests were used as post hoc tests. Control (PBS exposure) $n = 4$, PAE $n = 4$. *,**$p < 0.05$, 0.01. PE phosphatidylethanolamine. **c, d** There is no difference in abundance of fatty acid species between control and PAE in PC of the motor cortex at P30. Two-way ANOVA found no interaction between fatty acid species and exposure types. Control $n = 4$, PAE $n = 4$. Inset shows the magnified view of the fatty acid species that are less abundant. Graphs represent mean ± SEM. Each dot represents an individual animal. PC phosphatidylcholine.

of the fatty acids that are in between C26 and C36 are thought to be substrates for ELOVL4[45,56]. In the retina, PUFAs elongated by ELOVL4 are incorporated into PC[57]. Therefore, we measured PUFA between C26 and C36 in PC but also in PE, in which we observed changes in fatty acids in PAE cortex (Fig. 2b, d). The analysis revealed that only 26:6 PUFA, the shortest fatty acid substrate of ELOVL4 enzyme, was decreased in PE of PAE cortex (Fig. 2b). However, no difference was observed in any of the fatty acid molecules of PC between control and PAE mice (Fig. 2d). The reduction of ELOVL4 substrates was consistent with the increase of ELOVL4 in the cortex of PAE mice. However, unexpectedly, the increase of the very long chain fatty acids that ELOVL4 produces was not observed. This indicated a possibility that those very long chain fatty acids are metabolized immediately after the production by excessive ELOVL4.

**Accumulation of lipid droplets (LDs) in PAE brain.** The results described above suggested that ELOVL4-produced very long chain fatty acids are metabolized in PAE motor cortex, or potentially sequestered away to prevent lipid toxicity. LDs store such intracellular fatty acids by incorporating them into neutral lipids such as triglycerides, while excessive accumulation of LDs is one of the pathological signatures in neurodegenerative diseases and aging brains[58]. In addition, an increase in both palmitic acid and arachidonic acid, which showed an increase in PAE motor cortex (Fig. 2a), promotes the formation of LDs in hepatic cells and monocytes[59,60]. Altogether, we hypothesized that PAE might facilitate formation of LDs in the motor cortex.

To detect LDs, we used oil red o (ORO) which stains neutral triglycerides and lipids in the brain. We first confirmed ORO staining with a 12-month-old mouse as LDs accumulate in aging brains[61]. ORO-labeled LDs were found inside cells not only in the motor cortex but also in cingulate and piriform cortices, striatum, hippocampus, and lateral ventricle (LV) wall where periventricular glial cells are located (Supplementary Fig. 3), similar to observations made in aging studies[61,62].

We then examined LD accumulation in the motor cortex and several brain regions of PAE and control mice at P30, as depicted in Fig. 3a. The number of LD accumulating cells was increased in the motor cortex of PAE mice compared with that of control mice (Fig. 3b, c). Notably, an increase of LD included cells was also observed in other brain regions such as the striatum, CA3, and dentate gyrus of the hippocampus (Fig. 3b). Although statistically not significant, the piriform and cingulate cortices and LV wall showed trends of increase in the number of cells with ORO labeling. As an alternative approach, we stained brain slices with BODIPY, a fluorescent dye that stains neutral lipids and is commonly used to detect LDs[63], and found that NeuN positive cells colocalize with LDs in PAE brains, consistent with ORO staining (Supplementary Fig. 4). Although ORO staining did not give clear image inside cell, the BODIPY staining also showed multiple LDs with different sizes in a neuron in PAE brains.

Next, we examined the LD droplet formation during postnatal development. Given that the dam's milk contains a high level of fat at approximately 30%[64] and high fat induces LD accumulation

in periventricular glial cells adjacent to LV wall in mice[65], there was a possibility that the LD formation may be complicated around the weaning when the pups are no longer suckling. Thus, the number of ORO-positive cells in the brain was examined on three postnatal days, before weaning (P15), the day of weaning (P20), and after weaning (P25).

We found that the LV wall and the two regions of gray matter in the cortex showed different dynamics of LD depositions during normal development (Supplementary Fig. 5). In PAE, the dynamics were also different between the development of LV wall and the gray matter (Supplementary Fig. 5). Only at P20 (and P30 in Fig. 3b), but not at P15 or P25, the differential accumulations of LDs were observed between PAE and control cortices (Supplementary Fig. 5), suggesting spatiotemporal dynamics of LD formation around weaning date.

**The gut microbiome that are associated with fatty acid biogenesis and metabolism are altered by PAE.** Accumulation of LDs is one of the neuropathological hallmarks in neurodegenerative diseases such as PD and AD. These patients also show changes in both peripheral triglycerides levels[66,67] and gut microbiota composition[68,69], suggesting ingrained changes in lipid metabolism in these patients. Given that our PAE mice accumulated LDs in the brain and previous studies showed an association between PAE and metabolic disorders such as obesity and hyperlipidemia[6–8], we examined body weight, plasma triglyceride level, and blood glucose level in our animals. However, none of those metabolic measurements were altered in our acute PAE mouse model (Supplementary Fig. 6), indicating neurobehavioral problems without manifested signs of metabolic dysfunctions in PAE.

Then we examined if the gut microbiota composition is altered by PAE. As shown in the experimental timeline in Fig. 4a, following collections of fecal pellets for 16 S ribosomal RNA (rRNA) sequencing, animals were placed on an accelerated rotarod to assess their motor learning by conducting 3 trials per day for two consecutive days. The next day, animals were placed in an elevated plus maze (EPM) to assess anxiety. The behavioral test results showed that PAE group have both motor learning deficit and anxiety phenotype that were demonstrated by a reduction in the learning index and time spent in the open arm, respectively (Fig. 4b, c, d). In addition to reduced open arm time in EPM, the numbers of open arm entries and closed arm entries, as well as the total number of entries in arms were decreased in PAE group (Supplementary Fig. 7a, b, c). However, while there was no difference in the time spent in closed arms between control and PAE groups (Supplementary Fig. 7d), mice from the PAE group spent more time in the center (Supplementary Fig. 7e). The increased center time but a decreased open arm time in PAE mice indicated that the mice showed exploring behavior but avoid entering the open arm. Moderate but significant correlations between motor learning index and anxiety measurement, open arm time in EPM, were also found (Supplementary Fig. 8). In addition, the changes in these behaviors are less likely due to the locomotion issue, as the PAE model does not show any changes in the open field test compared to control mice in our

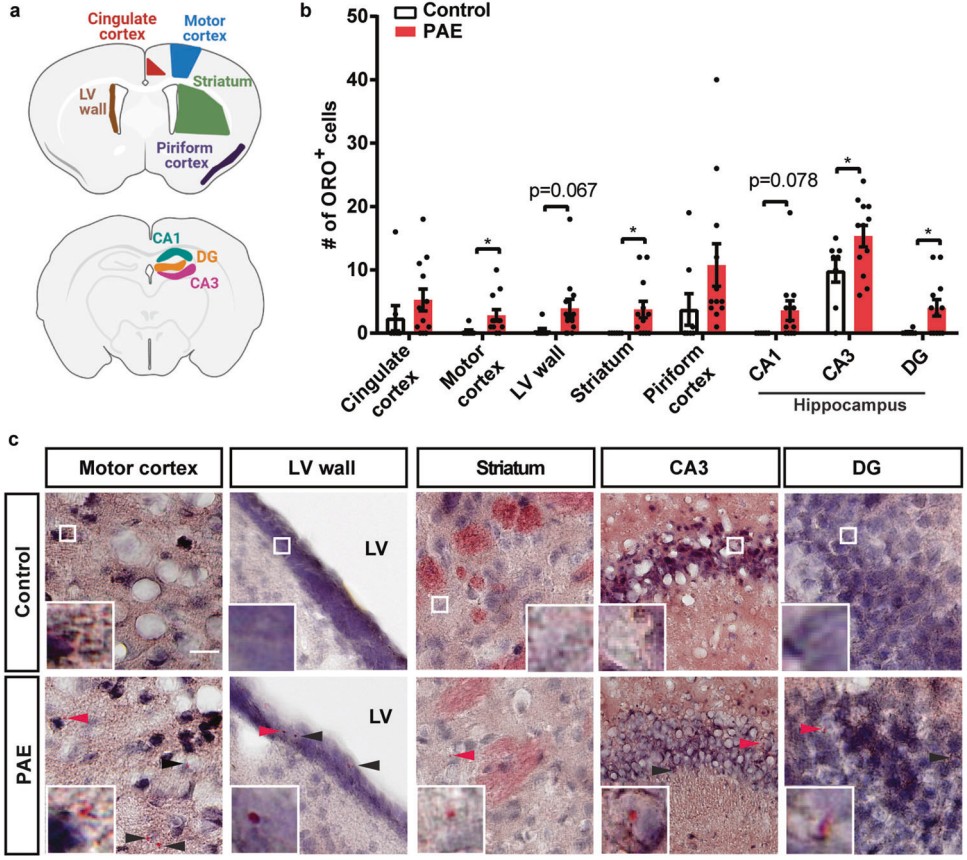

**Fig. 3 LDs are accumulated in various brain regions in PAE mice at P30. a** Highlighted areas in the schema indicate brain regions analyzed in mice. **b** The numbers of LD included cells were quantified in 200 × 200 μm² area in each brain region. The data show an increase of LD accumulating cells in the motor cortex ($p = 0.0352$), striatum ($p = 0.0311$), both DG and CA3 of the hippocampus ($p = 0.0267$ and $0.0466$, respectively), and lateral ventricle (LV) wall ($p = 0.067$) in PAE mice. *$p < 0.05$. Student's $t$-test. Control $n = 8$; PAE $n = 12$. Graphs represent mean ± SEM. Each dot represents an individual animal. **c** Representative images of ORO stained brain areas show changes in the number of cells accumulating LDs. Insets show higher magnification views around red arrowheads or white square of each panel. Scale bar = 20 μm. Arrowheads indicate LDs.

previous study[44]. Consistently, motor coordination evaluated at trial 1 does not show any differences between PAE and control either (Fig. 4c)[44].

To investigate changes in microbiome, we performed 16 s rRNA sequencing and analyzed using Mothur[70]. After alignment and mapping with the SILVA v138 reference database, the operational taxonomic unit (OTUs) were clustered at 97% identity threshold to sort reads into each gut bacteria. OTUs with less than 4 read counts were removed from each sample, and then bacteria with its OTU counts showing less than 10 % prevalence in entire samples were removed from the analysis. After those filtrations, the final numbers of the total read counts were similar between samples, ranging from 81765 to 82281 (Supplementary Fig. 9a). The plateau of rarefaction curves showed that most of the abundant species are included in each sample (Supplementary Fig. 9b). The curves of PAE samples indicated higher variability in the number of detected OTUs (Supplementary Fig. 9b).

As a result, 35 bacterial families were found in the samples; *Lachnospiraceae*, *Unclassified Bacteroidales*, and *Lactobacillaceae* were some of the highly abundant microbiota in the samples (Fig. 4e). The bacterial diversity analysis revealed that PAE mice have lower alpha diversity compared with control mice (Fig. 4f), suggesting decreased richness of bacterial composition in PAE mice compared to control mice. Furthermore, beta diversity based on Bray-Curtis distance matrix indicated a difference between control and PAE mice in microbial composition (Fig. 4f).

We further investigated differentially abundant microbiota at the genus level. Analysis revealed that 12 bacterial genera (8 increased and 4 decreased in PAE) were differentially abundant between PAE and control (Fig. 4g). To test the correlation between behavioral phenotypes and abundance of microbiota, we performed Spearman's correlation analysis between the abundance of significantly altered bacterial genera and animal's learning index in the accelerated rotarod test and time spent in the open arm of EPM in PAE group. The analysis revealed that the abundance of three microbiota, *Blautia*, *Unclassified Erysipelatoclostridiaceae*, and *Muribaculaceae_ge*, showed a positive correlation with the motor learning index, whereas *Unclassified Erysipelotrichales* showed a negative correlation with the index (Fig. 4h). *Blautia* is a gut microbial genus that produces SCFAs such as butyric acid and acetic acid[71]. *Muribaculaceae*, also known as *S24-7*, is a family of bacteria that produces the enzymes for carbohydrate degradation[72]. *Erysipelotoclostridiaceae* is a family of *Firmicutes* that are involved in carbohydrate metabolism[73]. With time spent in the open arm in EPM, *Muribaculaceae_ge* and *Unclassified Ruminococcaceae* showed positive and negative correlations, respectively (Fig. 4h). *Ruminococcaceae* degrades cellulose and hemicellulose and subsequently converts these compounds to SCFAs, which can be absorbed and used for generation of energy by the host[74].

Using TaxFun2, changes in functional profiles due to altered microbiota compositions were examined[75]. The data revealed that lipid metabolism is the most affected biological function by PAE (Supplementary Fig. 10). We further examined which lipid

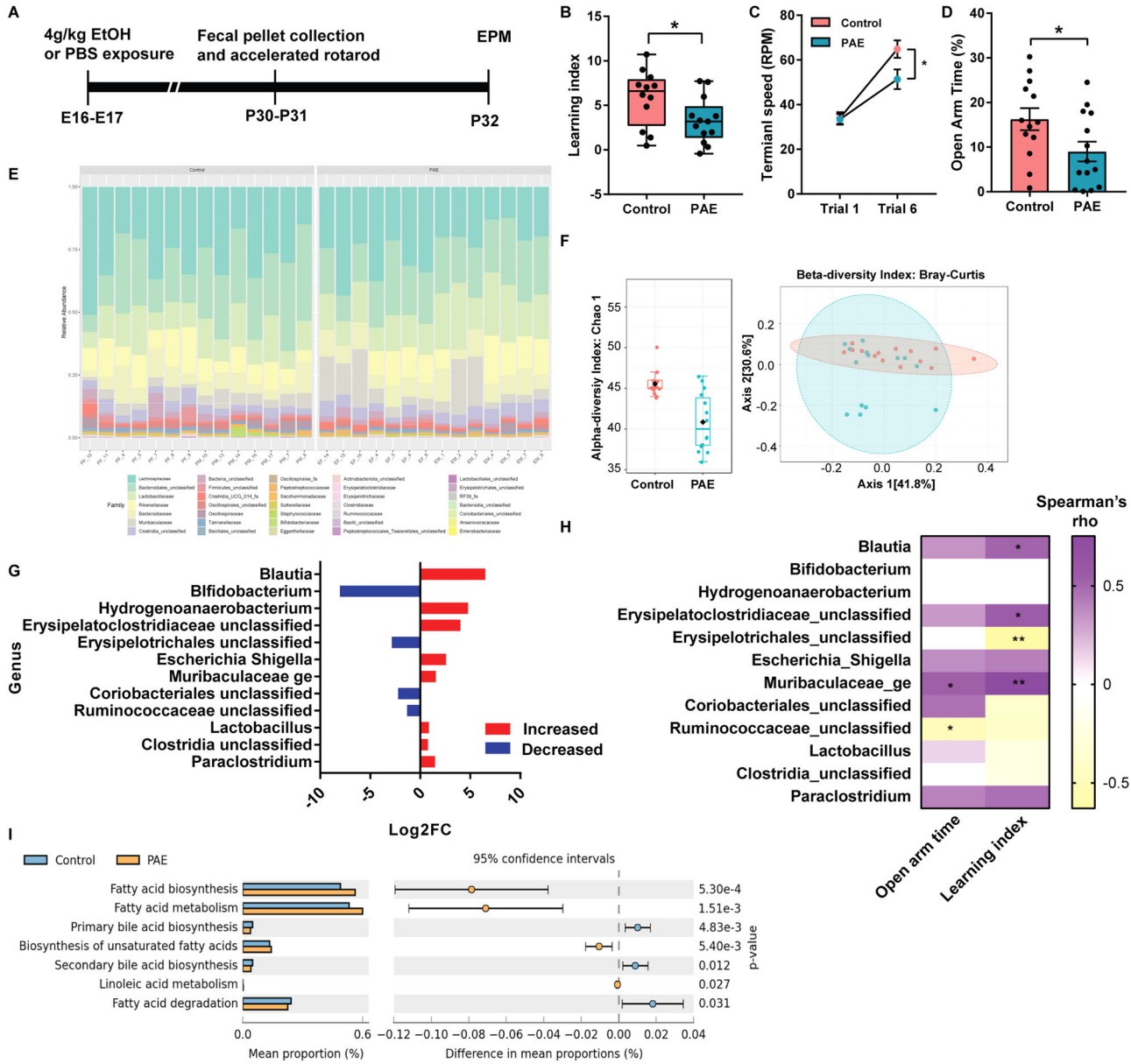

**Fig. 4 Gut dysbiosis is associated with behavioral problems in PAE mice. a** Timeline of the experiment. **b** The PAE mice score lower in the learning index in the accelerated rotarod test ($p = 0.0351$). Student's $t$-test. Control $n = 12$; PAE $n = 13$. Graph represents a box blot with 25th, median, and 75th percentiles, and whiskers extend to the lowest and highest values. Each dot represents an individual animal. **c** PAE mice reach lower terminal speed in the last trial (Trial 6) compared to control mice (adj.$p = 0.0257$). Two-way ANOVA revealed no interaction between trials and exposure. Tukey's post hoc test was used for multiple comparisons. Control $n = 12$; PAE $n = 13$. Each dot on the graph represents a group mean ± SEM. **d** PAE mice spend less time in the open arm compared to the control mice ($p = 0.0370$). Student's $t$-test. Control $n = 13$; PAE n = 14. **c, d** Graph represents mean ± SEM. Each dot represents an individual animal. **e** Stacked bar plots show the taxonomic composition of the microbial community at the family level in each animal used for analysis. **f** Microbial diversity analysis reveals that PAE mice have smaller alpha diversity compared to that of control mice ($p = 0.00030926$) as well as differential microbial communities (beta diversity). Alpha diversity was compared using Chao1 at the genus level. Graph represents a boxplot with 25th, median, and 75th percentiles. Whiskers represent 9 and 91 percentiles, and outside of whiskers represent outliers. Each dot represents an individual animal, and the black diamond represents the mean. Beta diversity analysis at the genus level shows that PAE and control mice have differential microbial communities. $F = 3.2946$, $R^2 = 0.11246$, $p = 0.017$ by Permutational ANOVA (PERMANOVA). Graph represents 2-D Principal Coordinate Analysis (PCoA). Each dot represents an individual animal. **g** Using EdgeR, 12 differentially abundant taxonomy are identified at the genus level. Graph represents log2 Fold Change (FC), and bacteria is listed in the order of the most significant (*Blautia*) to the least significant (*Paraclostridium*). Red bar increased in PAE, Blue bar decreased in PAE. **h** The abundance of bacterial taxa at the genus level in PAE mice correlates with levels of motor learning and anxiety. Color of heatmap indicates rho values by Spearman's correlation analysis. **i** Functional profiles of the microbiome predict changes in fatty acid-related pathways in PAE mice. Graph represents the difference in mean proportions (%) with 95% confidence intervals calculated by Welch's inverted method. *,**$p < 0.05$, 0.01.

metabolism-related pathways are altered in PAE mice and found that biosynthesis and metabolism of fatty acids and biosynthesis of unsaturated fatty acids are increased. Metabolism of linoleic acid, a PUFA that is a substrate to be elongated to the arachidonic acid[76], was also increased (Fig. 4i). On the other hand, fatty acid degradation was decreased in PAE mice (Fig. 4i). Notably, the analysis also revealed that both primary and secondary bile acid biosynthesis is decreased in PAE mice (Fig. 4i). Bile acids help breaking down fats into fatty acids to facilitate their absorption into cells[77]. The liver synthesizes primary bile acids from which secondary bile acids are transformed by gut bacteria[78]. Collectively, these results suggested that the fatty acid biosynthesis is increased while degradation is decreased in PAE mice, resulting in excessive depositions of fatty acids by the gut microbiota in PAE mice.

***Ruminococcaceae* show the strongest correlation with LD accumulation in the brain and behavioral phenotypes**. To comprehensively examine the correlations between obtained biological measures, we integrated data using a mixOmics-based R package called DIABLO[79,80]. Different from Spearman's correlation analysis above, which used only PAE samples (Fig. 4h), DIABLO algorithm enables the integration of all PAE and control samples to identify key features across different data types while discriminating between multiple phenotypic groups. The neurobehavioral phenotypes, brain LD, and microbiome results obtained from an animal in either PAE or control group were used for the analysis. As shown in Fig. 5, the motor learning index showed a strong negative correlation with the total brain LD accumulations. Among the brain regions, the piriform cortex was one of the brain regions we saw a strong correlation with anxiety measures. The piriform cortex is mainly known for the odor processing region, but it also receives input from the basolateral amygdala, a brain region important for anxiety, to form a cortical circuit to shape responses to the threatening stimuli[81].

There were stronger correlations with LD accumulation and neurobehavioral measures in the microbiota that are known to be

associated with fatty acid synthesis and metabolisms, such as *Unclassified Ruminococcaceae*[74], *Blautia*[71], as well as carbohydrate metabolism associated microbiota including *Muribaculaceae_ge*[72] and *Unclassified Erysipelatoclostridiaceae*[73]. In humans, consuming *Escherichia Shigella* contaminated food or water can cause serious illness[82]. Similarly, in mice, intraperitoneal administration of *Shigella* bacteria induced severe diarrhea and acute inflammation, similar to humans[83]. In addition, *Escherichia Shigella* was shown to be increased in anxiety disorder patients[84]. Among all detected bacteria, *Unclassified Ruminococcaceae* showed the strongest negative correlation with the total number of accumulated LDs, but a positive correlation with the motor learning index, suggesting that *Ruminococcaceae* may affect the behavior through the impact on the lipid metabolism in the brain.

## Discussion

Increased expression of proteins encoding enzymes that are involved in fatty acid biogenesis and modification in the cortical neurons in PAE mice long after prenatal exposure (Fig. 1a and previously published work[44]) led us to investigate the fatty acid composition in membrane phospholipid in the cortex (Fig. 2). We also applied MRM to quantify the very long chain fatty acid species in PAE and FASD research for the first time. Despite the increased expression of ELOVL4 in the cortex (Fig. 1a), an enzyme that elongates the C26, there were no changes in C28 and the longer chain fatty acids (Fig. 2b). However, the reduction of C26:6 in PE (Fig. 2b) suggested a possibility that, despite increased usage of C26:6 by ELOVL4, the produced very long chain fatty acids are broken down immediately after the elongation; therefore, no increase was observed in phospholipid.

We also found increases in palmitic and arachidonic acids in PE in PAE mice (Fig. 2a). Of note, previous studies showed increased arachidonic acids in maternal plasma from mothers who consumed alcohol during pregnancy and had children with lower cognitive test scores compared to those mothers who consumed alcohol but had children with the normal scores[19]. In the same study, palmitic acids in maternal plasma were also positively correlated with alcohol intake.

LD biogenesis begins in the endoplasmic reticulum (ER) bilayer. Overloading of excessive lipid entered into ER triggers the formation of LDs[85]. No changes in the amount of the fatty acids that are longer than C26 and simultaneous reduction of C26 in the plasma membrane of the PAE brain cells (Fig. 2) indicated a possibility that excessively elongated very long chain fatty acids are being metabolized and sequestered by forming LDs to prevent lipidopathy. In fact, we observed the increase of LDs in various brain regions in PAE mice (Fig. 3). However, the distribution of brain LDs was sparse and smaller in size in P30 mice (Fig. 3, Supplementary Fig. 4) compared to 1 year old mouse (Supplementary Fig. 3)[86,87]. These differences could be due to the young age (P30) when brain LDs were examined, as previous studies have reported alteration in brain lipid metabolism and composition with aging, which could contribute to brain LD formation and dynamics[58,88].

Another possible explanation as to why there was no increase of C26 observed despite the increase of ELOVL4 in the PAE cortex might be due to the specific subcellular localization of these very long chain fatty acids: such as synaptic vesicles[89] while we collected PC and PE of all membrane fractions that potentially mask the differences in the synaptic compartment.

Our study revealed the accumulation of the LDs in the brains of PAE animals (Fig. 3) that show various cognitive and behavioral problems[44,46]. There were slight differences in the levels of accumulation between brain regions; however, all of the brain regions showed an increase of LDs in PAE, suggesting brain-wide

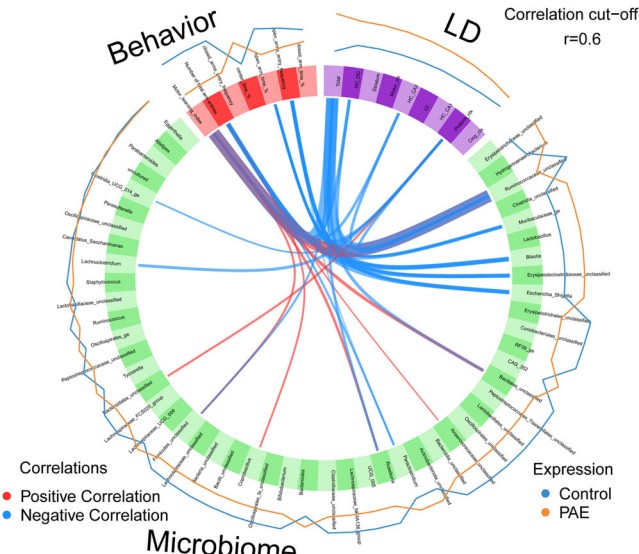

**Fig. 5 *Ruminococcaceae* is strongly correlated with brain LD accumulation and behavioral phenotype.** Circos plot shows positive and negative correlations between the measures obtained in behavioral tests (accelerated rotarod and EPM), number of LD accumulating cells in the brain, and microbial abundance. The thickness of the link corresponds to the correlation coefficient (r). Control n = 8, PAE n = 12. HC hippocampus, DG dentate gyrus, LV lateral ventricle wall.

issues. The variability in LD accumulation levels could be due to the difference in lipid composition among different brain regions, where the prefrontal cortex and motor cortex show similar lipid composition in the membrane but are different from other brain regions such as the hippocampus, striatum, and cerebellum[90]. In addition, correlations were also observed between behavioral measurements and LD accumulations in brain regions known to be involved in motor learning and anxiety (Fig. 5). Interestingly, the LDs appeared as transient pathological features that are likely to be affected by diet during postnatal development. In our observation, the accumulation of LDs was dynamic during the weaning time (Supplementary Fig. 5).

In the functional annotation to changes in the microbial profiles in PAE animals, we found an increase in the fatty acid synthesis pathway but a reduction in fatty acid degradation, suggesting a possible systemic increase in amounts of fatty acids in PAE mice (Fig. 4i). This prediction was consistent with the lipid-related brain pathology and changes in fatty acid contents in membrane phospholipids in the brain (Fig. 2). Further functional analysis of microbiome revealed that the metabolic pathway of linoleic acid, which is a precursor fatty acid for arachidonic acid[76], was also increased in PAE mice (Fig. 4i). This change might be associated with the increase of the arachidonic acid in PAE brains (Fig. 2a). The primary and secondary bile acid synthesis were also predicted to be changed in PAE's microbiome profiles (Fig. 4i). This finding was interesting because the changes in bile acid biogenesis were observed from the biosignature of microbiome-derived metabolites in both alcohol-exposed dams and their fetuses[41]. Collectively, these biosignatures predicted from changes in gut microbial compositions suggest that the systemic changes in fatty acid biogenesis and metabolism and that those may be linked to the changes in PAE brains.

Association analysis between neurobehavior, lipid-associated brain pathology, and gut microbiome revealed the strong correlations of microbiota that are involved in fatty-acid biogenesis and metabolism with brain pathology and the behavior (Fig. 5). Among those microorganisms, Unclassified Ruminococcaceae showed the strongest associations with both brain pathology and behavior. Ruminococcaceae are found in both human and mouse gut microbiome, but the abundance is higher in humans than in mice[91]. Interestingly, the abundance of Ruminococcaceae in the gut is associated in both positive and negative directions with neurobehavioral measures, depending on which health measures and which brain functions were used[92–94]. For example, Ruminococcaceae was associated with lower anger, greater cognitive functions, and varied personality traits in humans[92]. In addition, AD patients showed a reduced abundance of Ruminococcaceae[93]. On the other hand, autistic patients showed higher abundance compared with the control subjects[94]. Furthermore, hepatic encephalopathy patients treated with an oral capsular fecal microbial transplant that includes enriched Ruminococcaceae showed improvement in cognitive problems[95]. Therefore, Ruminococcaceae may serve as a biomarker or/and become a probiotic treatment option for the FASD. The integrative analysis (Fig. 5) also showed an intriguing possibility that the gut microbiota might directly affect the lipid pathogenesis without changing metabolism in the entire body at a significant level. An exciting hypothetical mechanism is that metabolites of gut microbiota affect vagus nerve signaling[96] and that vagus nerve stimulation directly affects lipid composition in various brain regions, including the striatum and motor cortex[97], which could be mediated by the gut-brain axis[34]. As a possible molecular mechanism that controls LD formation in the brain by gut microbiota in PAE animal, SCFAs that are produced by gut microbiota may inhibit histone deacetylase function[30,31], consequently modifying gene expression, including genes like Fasn and Elovl4, within the brain.

Although changes in systemic metabolic measures such as blood glucose level and body weight have been reported in chronic alcohol exposure animal models[98,99] and adult FASD patients[8], those measures were not altered by acute prenatal alcohol exposure in our animal model (Supplementary Fig. 6), suggesting no relation with neurobehavioral problems or lipid-associated brain pathology. This may be due to milder phenotypes after acute PAE compared to a chronic PAE model. However, importantly, our model exhibits various neurobehavioral problems, including motor learning deficits and anxiety, and disruption in fatty acid related metabolism in brain and gut. Therefore, our results suggest that the effects of gut microbiota may profoundly and directly affect the brain pathology and function without affecting systemic metabolism in PAE.

## Methods

**Animals**. All animal experiments were conducted in accordance with the National Institutes of Health Guide for the Care and Use of Laboratory Animals and approved by the Institutional Animal Care and Use Committee at Children's National Hospital (Protocol no. 00030323). To expose pregnant dam to alcohol, 4.0 g/kg body weight of ethanol (PAE group) or PBS (control group) was injected intraperitoneally (i.p.) at E16 and E17 to timed pregnant CD-1 (strain code: 022) mice that were purchased from Charles River Laboratories (Wilmington, MA) as previously done[44]. The dams were randomly assigned to PAE or control group without any predetermined criteria. The day of birth was designated as postnatal day (P) 0. For all experiments, a similar or equal number of female and male mice around the age of P30 were used, unless otherwise specified in the figure legends. All the animals were maintained on a light-dark cycle (lights on 6:00–18:00) at a constant temperature (22 ± 1°C).

**Accelerated rotarod**. The test was performed as previously described[44]. Briefly, the mice were placed on a rotating bar, and the length of time that they could retain their balance during acceleration of rotation to a max speed of 80 rpm in 5 min was recorded. The testing phase consisted of 2 consecutive days with three trials per day. Each trial was at least 15 min apart and was terminated when the mouse fell off, made one complete rotation without walking on the rotating rod, or reached maximum speed after the 5 min session. The motor learning index was calculated by averaging the difference in terminal speed of the two consecutive trials.

**Elevated plus maze test (EPM)**. The maze is a grey plus-shaped apparatus with two open arms and two closed arms linked by a central platform. Mice were individually put in the center of the maze facing an open arm and allowed to explore the maze for 300 s. A video was recorded during the experiment and analyzed with MouBeat ImageJ Plugin as per the user guide as previously done[46,100].

**Immunohistochemistry**. Mice were deeply anesthetized with isoflurane (Henry Schein, Melville, NY) and perfused transcardially with 10 ml of ice-cold PBS followed by 10 ml of chilled 4% paraformaldehyde (PFA). The brains were removed and immerse-fixed in 4% PFA at 4 °C overnight. Then incubated in 10% and 30% sucrose in PBS for 24 h sequentially at 4 °C and embedded in the OCT compound (cat# 4583; VWR, Randor, PA). Coronal sections were cut at 20 or 50 μm on a cryostat (CM3050S; Leica, Buffalo Grove, IL).

Free-floating staining was performed with 50 μm thick brain sections. Briefly, antigen retrieval was performed following the manufacturer's protocol (cat# 00-4955-58; ThermoFisher, Waltham, MA) when necessary. Then sections were incubated in hydrogen peroxide in methanol (1:4) solution for 30 min at −20 °C to inactivate endogenous peroxidase activity. After rinsing with PBS containing 0.01% of Tween-20 (PBS-T), sections were incubated with 2% BSA for 30 min at room temperature for blocking. Sections were then incubated with ELOVL4 (1:500 gifted from Dr. Robert Anderson or 1:200; cat# 224608; Abcam, Waltham, MA), and/or NeuN (1:300; cat# MAB377; EMD Millipore, Burlington, MA) primary antibodies for overnight at 4 °C. For ELOVL4 immunolabeling, 2 h incubation with biotinylated anti-rabbit IgG (cat# 711-065-152; Jackson ImmunoResearch, West Grove, PA) diluted at 1:300 followed by 1 h incubation with avidin-biotin complex (1:1:100; cat# 32020; ThermoFisher) and 1 h incubation with TSA plus Cyanine-3 (1:300; cat# NEL744001KT; Akoya Biosciences, Marlborough, MA) was performed. For NeuN staining, sections were incubated for 2 h with HRP-conjugated anti-mouse IgG diluted at 1:300 and followed by 1 h incubation with TSA plus Cyanine-2 (1:300; cat#NEL745001KT, Akoya Biosciences). Sections were counterstained with DAPI and mounted on slides with CC/Mount mounting medium (cat# C9368; Sigma, St. Louis, MO). Images were acquired using a confocal microscope (FV1000; Olympus Center Valley, PA). All images were analyzed with ImageJ using the Cell Counter tool. In the same PAE model, we previously showed no change in the

density of cell number in PAE brain[44]. Thus, positive cell counts/area is valid method for comparing the number of cells that express a certain molecule.

**Oil Red O (ORO) staining**. 20 μm thick brain sections were prepared as described above. For staining, ORO stock solution was prepared by dissolving 0.05 g of ORO powder (cat#O0625; Sigma) in 10 mL of isopropanol. Then 60% ORO working solution was prepared freshly with distilled water and filtered before use. Brain slices were incubated in 60% ORO solution for 10 min, washed thoroughly with distilled water, and incubated for 15 min with hematoxylin for counterstaining. Then the slices were mounted on slides with CC/Mount mounting medium, and images were acquired with an Olympus VS120 microscope. Brightness and contrast were adjusted with CellSens, and ORO-positive cells were manually counted with ImageJ Cell Counter tool.

**BODIPY staining**. 50 μm thick brain sections were prepared as described above. The brain sections were washed in PBS then incubated for 10 min in 2% BSA for blocking, Then incubated in NeuN primary antibody (1:500; cat# MAB377; EMD Millipore) overnight at 4 °C. The next day, brain sections were washed in PBS and probed with Cy3-conjugated anti-mouse IgG (1:100; cat# 115-165-003; Jackson ImmunoResearch) for 3 h. Then incubated in BODIPY 493/503 (cat# D3922; ThermoFisher) for 15 min, followed by DAPI for counterstaining. Images were acquired using a confocal microscope (FV1000).

**Lipidomics of phospholipid fatty acids**. Motor cortical regions were dissected from both hemispheres of P30 mice and snap-frozen in liquid nitrogen. Samples were stored at −80 °C and shipped to Lipid Analysis Core at Emory University for phospholipid fatty acid quantification. Lipids were extracted from the cortical tissues using Bligh and Dyer method[101], and extracted lipids were directly loaded onto the mass spectrometer (SCIENX QTRAP LC-MS/MS system; Framingham, MA) for targeted lipidomics. The phospholipid class was selectively targeted using characteristic scans, and the same mass spectrometer was operated in multi reaction monitoring (MRM) mode to detect specific species of very long chain fatty acids that are presumably synthesized by ELOVL4[102]. Detected fatty acids are converted to relative percentages by dividing the peak intensity of each fatty acid species by the total peak intensities detected per animal. To compare total PC and PE contents, all peak intensities from an individual animal were summed and divided by the total weight of the tissue.

**Plasma collection and measurement of triglycerides**. Whole blood was collected from non-fasted and overnight fasted (16 h) mice in an EDTA-treated tube (cat# 365974; BD, Franklin Lakes, NJ) by cardiac puncture with a 23–25 G needle from a deeply anesthetized animal. Cells were removed from the plasma by centrifugation at 2000 x g at 4 °C for 10 min. The supernatant was carefully transferred to a clean centrifuge tube by pipetting and stored in a −80 °C freezer until further analysis. Measurement of triglycerides was carried out by a commercially available kit (cat# ab65336; Abcam) that detects glycerol levels after converting triglycerides to free fatty acids and glycerol by following the manufacturer's protocol. Absorbance was measured with xMARK microplate spectrophotometer (cat# 1681150; Bio-rad, Hercules, CA).

**Fecal pellet collection and the 16 S rRNA sequencing**. Fecal pellets were collected from P30 animals and put on dry ice immediately after collection. DNA was extracted using the MagMAXTM CORE Nucleic Acid Purification Kit (Thermo Fisher Scientific, Waltham, MA), according to manufacturer instructions, and subjected to the library preparation for 16 s rRNA V3-V4 region sequencing by using repliQa HiFi ToughMix (Quantabio, Beverly, MA). Next-generation sequencing was performed on an Illumina MiSeq by 2 x 300 bp paired-end readings. Mothur[70] was used to align and map sequences with the SILVA v138 database. Chimeric reads were removed with VSEARCH, and operational taxonomic units (OTU) were clustered at 97% similarity. Biom files generated by Mothur were put into MicrobiomeAnalyst (https://www.microbiomeanalyst.ca/) for downstream analysis after removing low counts less than 4 and prevalence in samples less than 10%. Filtered data were scaled by total sum scaling, and alpha and beta diversity were determined by Chao1 and Bray-Curtis Index, respectively. Differential abundance analysis was carried out using EdgeR. To perform functional profiles analysis, representative OTU FASTA sequences and OTU count tables were retrieved from Mothur and analyzed with "Tax4fun2" package in R[75]. Statistical Analysis of taxonomic and functional profiles (STAMP)[103] was used to perform statistical analysis to compare metagenomic functional profiles between control and PAE mice.

**Multi-omics integrative correlation analysis**. DIABLO multi-omics integration method from the mixOmics R package[79] was applied to integrate different types of datasets from the same animals that have measurements in all three data sets: microbiome, data of the numbers of cells with lipid droplet (LD) in the brain regions, and results of behavioral tests (accelerated rotarod and EPM tests). We found 8 control and 12 PAE mice that meet this criterion. Centered log ratio (Clr) transformed microbiome and Z-score transformed the number of LDs accumulated

in the brain, and behavioral measurements were used as the input data. The data integration was carried out following the manual. Briefly, a matrix model was designed to connect all datasets with a link strength at 1 between two components were used to generate the final DIABLO model. The circos plot shows the correlation strength and directionality between variables of different types of our datasets with a correlation coefficient cut off $>|0.6|$.

**Statistics and reproducibility**. All histological data were acquired from the defined subregions of the brain. All groups consisted of mice from at least two different litters with a similar number of female and male mice otherwise noted in the figure legends. The data from males and females combined, unless there was a statistically significant difference between the sexes, as indicated in the figure and figure legends. The number of n indicated in the figure legends is that of biological replicates. The sample size for each experiment was determined based on previous experience in similar experiments. For all in vivo experiments, no animals were excluded from the analysis. Fatty acid analysis (Fig. 2) and 16 s RNA sequencing (Fig. 4) were analyzed by the personnel who were group-blinded. Behavioral experiments were performed unblinded; however, automated analysis was used for EPM analysis (Fig. 4). All the immunohistological analysis was done group-blinded by the investigator. Plasma triglycerides, blood glucose, and body weight measurements were performed unblinded (Supplementary Fig. 6). All of the statistical analysis was carried out with GraphPad Prism 7.01. We performed the D'Agostino–Pearson to test the normality of data. For the data that passed the normality test, Student's t-tests or one-way or two-way ANOVA was used. Post hoc Tukey's or Bonferroni's test was done as described in the figure legends. Simple main effects were reported when there was a statistically significant interaction between independent variables by two-way ANOVA. Pearson's or Spearman's correlation coefficient calculation was done for normally or non-normally distributed data, respectively. P values of less than 0.05 were considered statistically significant. Source data for graphs and charts are presented in Supplementary data 1.

**Reporting summary**. Further information on research design is available in the Nature Portfolio Reporting Summary linked to this article.

## Data availability
All original 16 s rRNA sequencing raw data have been deposited to the National Library of Medicine Bioproject under accession ID PRJNA842719. Source data for graphs and charts in the article can be found in Supplementary data 1. All other data in this article are available from the corresponding author on reasonable request.

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

## Acknowledgements

We would like to thank Dr. Robert E. Anderson for kindly providing ELOVL4 primary antibody. The 16 S rRNA sequencing work was processed at the Genome Sciences and Bioinformatics Facility at the Penn State University College of Medicine. The lipid analysis was supported in part by the Emory Integrated Lipidomics Core (EILC), which is subsidized by the Emory University School of Medicine and is one of the Emory Integrated Core Facilities. This study was supported by F31AA027693 (H.M.H), R01AA025215, R01AA026272 (K.H-T), and 1U54HD090257-01 from the NIH, District of Columbia Intellectual and Developmental Disabilities Research Center Award (DC-IDDRC) program. Additional support was provided by the Georgia Clinical & Translational Science Alliance of the National Institutes of Health under Award Number UL1TR002378. The content is solely the responsibility of the authors and does not necessarily reflect the official views of the National Institutes of Health.

## Author contributions

Conceptualization: H.M.H., K.H-T. Methodology: H.M.H., Y.I.K., A.B., S.M., M.I., K.H-T. Formal analysis: H.M.H., Y.I.K., A.B., S.M.; Investigation: H.M.H. K.H-T. Writing–review and editing: H.M.H., Y.I.K., K.H-T. Funding acquisition: H.M.H., K.H-T.; Supervision: K.H-T.

## Competing interests

The authors declare no competing interests.
