## [Peer Review File · Communications Biology]

Reviewers' comments:

Reviewer #1 (Remarks to the Author):

This is well written manuscript which reports the novel finding that gene expressions linked to fatty acid (FA) metabolism in the cerebral cortex of the PAE mouse model are altered. Specifically, palmitic acid and arachidonic acid in phospholipid composition of the cerebral cortex of PAE at postnatal day 30 mice were affected. They found a decrease of 26:6 fatty acid which was consistent with the increase of the enzyme which uses 26:6 as a substrate. They report lipid droplets (LDs) accumulated in the PAE brain, suggesting malfunctioning FA metabolism. FA-associated metabolic measurements were not affected by PAE, but the abundance of FA-related gut microbiota was altered including Ruminococcaceae that produces short chain fatty acid and is connected to LD formation. However, the evidence for the lipid droplets was not high enough resolution to support the claims. Several other missing quantifications of cell number reduce enthusiasm for the conclusions. Suggest major revisions prior to publication.

Major revisions needed:

Need a full discussion of lipid droplet formation in the brain but how microbiota in the gut can affect brain lipid deposition.

ABSTRACT

Revision

Need to specify the 'producing enzymes' in the sentences: "increase of palmitic acid was consistent with increase of the producing enzyme." and "Decrease of 26:6 FA was also consistent with the increase of the enzyme which uses 26:6 as a substrate for making very long chain FAs."

PAGE 3

Need to remove:

"collectively indicate that disturbed fatty acid metabolism can be the treatment target for FASD"
Suggested to replace with "collectivly support targeting disturbed fatty acid metabolism for potential to mitigate lipid dyshomeostasis in FASD."

PAGE 4

Need more detailed discussion of how "SCFA also affects neurobehavior through the gut-brain axis"

Change "body weights" to body weight

PAGE 5

Increase mRNA which is based on independent cell mRNA content is not comparable to increased cell number of ELOVL4 expressing cells.

" increase at the RNA level 38, the number of cells that express ELOVL4 proteins was significantly increased in the PAE primary motor cortex (Figure 1a, b"

Supplementary Figure The number of ELOVL4 expressing cells is increased in the cingulate cortex but not in the cerebellum. Should be included in the main figures since it is a major discussion point.

Counts were not normalized to the number of nuclei. There could be an increased number of cell bodies in the cingulate cortex, not just ELOVL4 expressing cells.

What other brain areas were investigated for ELOVL4.

Intracellular distribution is commonly shown by confocal microscopy, the current resolution especially not by Figure 1d where the scale bar is 50 microns. It is not possible to conclude "intracellular distribution pattern of ELOVL4 in the neurons was not altered by PAE (Figure 1b, Supplementary Figure 1b, d)"

PAGE 6

Fig 2 only shows PE, but PC is also discussed on page 6. PC species should also be included in Figure 2. A trend increase in PC 16:0 /20:4 should be discussed. The ratio PE/PC should be investigated by each species.

PAGE 7

Add "s"

the increase of the very long chain fatty acids that ELOVL4 produce"s" was not observed.

PAGE 7

"only in the motor cortex but also in cingulate and piriform cortices, striatum, hippocampus, and lateral ventricle (LV) wall where periventricular glial cells are located (Supplementary Figure 4), similar to observations made in aging studies"

All these brain regions should also be used for determining the number of ELOVL4 expressing cells and reported.

PAGE 7 – 8

Discussion of ORO is incomplete regarding the what is seen in the figures 3 and Supplementary Fig 4.

Figure 3, ORO LD are difficult to resolve in current magnification of Fig. 3. No magnification insets are provided for Control tissue staining. Some lipid droplets may be visible in CA3 and striatum.

Supplementary Figure 4. Validation of the ORO staining using aging mouse brain ORO staining is difficult to resolve at currently presented magnification in supplemental Figure 4.

Supplementary Figure 5. BODIPY signal should be quantified.

Supplementary Figure 6. Lipid droplets cannot be resolved at the magnification presented.

PAGE 9

"both motor learning deficit and anxiety phenotype that were demonstrated by a significant reduction in the learning index and time spent in the open arm, respectively (Figure 4b, c, d)."

However, in the EPM test, it is not possible to determine if there is a motor deficit or anxiety phenotype. If there is a motor deficit, an anxiety phenotype cannot be accurately determined. Because there is a decreased total number of arm entries, (open and closed), the motor deficit precludes any conclusions to be made about anxiety behaviors. Additional information about the motor deficit is required to make any conclusions about the anxiety behavior. Roto-rod is used to inform on the motor activities, not the learning behavior (PAGE 16). This indicates there is a motor deficit and the learning and anxiety behaviors cannot be determined.

Reviewer #2 (Remarks to the Author):

In this manuscript, Hwang et al. show that prenatal exposure to the alcohol (PEA) will result in the postnatal changes in contributors to the fatty acid metabolism. For example, authors found upregulation of very long chain fatty acids protein 4 (Elovl4) and fatty acid 93 synthase (Fasn), and also of palmitic and arachidonic acid in the neocortical wall. Overall, authors elegantly show that PEA results in disrupted fatty acid metabolism in postnatal brain. Furthermore, authors report PEA induced accumulation of lipid droplets (LD) and changes in the gut microbiota and in studied motor and anxiety behaviors (e.g. accelerated rotarod test, and learning index). Overall, this is a strong study with an elegant experimental design that allowed accumulation of data and strong statistics that support main conclusions.

This is well written manuscript which reports the novel finding that gene expressions linked to fatty acid (FA) metabolism in the cerebral cortex of the PAE mouse model are altered. Specifically, palmitic acid and arachidonic acid in phospholipid composition of the cerebral cortex of PAE at postnatal day 30 mice were affected. They found a decrease of 26:6 fatty acid which was consistent with the increase of the enzyme which uses 26:6 as a substrate. They report lipid droplets (LDs) accumulated in the PAE brain, suggesting malfunctioning FA metabolism. FA-associated metabolic measurements were not affected by PAE, but the abundance of FA-related gut microbiota was altered including Ruminococcaceae that produces short chain fatty acid and is connected to LD formation.

However, the evidence for the lipid droplets was not high enough resolution to support the claims. Several other missing quantifications of cell number reduce enthusiasm for the conclusions. Suggest major revisions prior to publication.

We sincerely appreciate the reviewer for dedicating their time to thoroughly review our work and for offering valuable, detailed suggestions. We improved the manuscript by addressing all points listed below.

Major revisions needed:

1. Need a full discussion of lipid droplet formation in the brain but how microbiota in the gut can affect brain lipid deposition.

We revised 3rd and 6th paragraphs in discussion as follows:

Added in line 297> LD biogenesis begins in the endoplasmic reticulum (ER) bilayer. Overloading of excessive lipid entered into ER triggers the formation of LDs⁸⁵.

Added in line 353> As a possible molecular mechanism that control LD formation in the brain by gut microbiota in PAE animal, SCFAs that are produced by gut microbiota may inhibit histone deacetylase function^{30,31}, consequently modifying gene expression, including genes like *Fasn* and *Elovl4*, within the brain.

2.ABSTRACT

Revision

Need to specify the 'producing enzymes' in the sentences: "increase of palmitic acid was consistent with increase of the producing enzyme." and "Decrease of 26:6 FA was also consistent with the increase of the enzyme which uses 26:6 as a substrate for making very long chain FAs."

Those enzymes are now specified in lines 32 and 34 in Abstract.

3.PAGE 3

Need to remove:

"collectively indicate that disturbed fatty acid metabolism can be the treatment target for FASD"

Suggested to replace with "collectively support targeting disturbed fatty acid metabolism for potential to mitigate lipid dyshomeostasis in FASD.

Changed as suggested in line 63.

4.PAGE 4

Need more detailed discussion of how “SCFA also affects neurobehavior through the gut-brain axis”

Added the following sentence: Line 68> SCFA also affects neurobehavior through the gut-brain axis^{27,28} by regulating the biogenesis, oxidation, and metabolisms of fatty acids in various tissues, likely in context dependent manner²⁹. Some types of SCFAs inhibit the functions of a histone deacetylase to improve memory and learning in normal animals^{30,31} and an animal model of meningitis³², while others lead to autism-like behavior issues^{33,34}.

5.Change “body weights” to body weight

Changed as suggested.

6.PAGE 5

Increase mRNA which is based on independent cell mRNA content is not comparable to increased cell number of ELOVL4 expressing cells.

“ increase at the RNA level 38, the number of cells that express ELOVL4 proteins was significantly increased in the PAE primary motor cortex (Figure 1a, b”

Reviewer's concern is valid. We changed the sentence as follows: Line 102> Similar to the increase at the RNA level⁴⁴, the number of cells that express ELOVL4 proteins was significantly increased in the PAE primary motor cortex (Figure 1a, b).

7.Supplementary Figure The number of ELOVL4 expressing cells is increased in the cingulate cortex but not in the cerebellum. Should be included in the main figures since it is a major discussion point. Counts were not normalized to the number of nuclei. There could be an increased number of cell bodies in the cingulate cortex, not just ELOVL4 expressing cells.

We moved the supplementary figure 1 to main figure 1 accordingly. We added a description citing our previous paper as follows; Line 96> in the same PAE model, The offspring, a model of FASD, does not have obvious morphological abnormality or changes in the density of neurons in the brain. Thus, positive cell counts/area is valid method for examination of the number of cells that express a certain molecule. We also added similar description in Materials and Methods.

8.What other brain areas were investigated for ELOVL4.

We added the data in new Supplementary Figure 1.

9.Intracellular distribution is commonly shown by confocal microscopy, the current resolution especially not by Figure 1d where the scale bar is 50 microns. It is not possible to conclude “intracellular distribution pattern of ELOVL4 in the neurons was not altered by PAE (Figure 1b, Supplementary Figure 1b, d)”

We modified the description to more accurately reflect our actual observations: Line 111> Widespread distribution of ELOVL4 in the cytoplasm is similar between PAE and the control.

10.PAGE 6

Fig 2 only shows PE, but PC is also discussed on page 6. PC species should also be included in Figure 2. A trend increase in PC 16:0 /20:4 should be discussed. The ratio PE/PC should be investigated by each species.

Following the reviewer's suggestion, PC data has been moved to Fig.2 from supplement. A description of PC16:0/20:4 data is now included in results (line 135). Given significant number of species is not detected in either PC or PE, we decided to compare the ratio using total phospholipids by following previous papers such as Lee et al., Scientific Reports 2018 and Montgomery et al., Scientific Reports 2017. We were also not able to find papers that compared PC/PE ratio in each species from brain tissues, and thus we felt lack of information that makes a foundation (i.e. for comparison of our control data with previous reports) for examining the effects of PAE.

11.PAGE 7

Add "s"

the increase of the very long chain fatty acids that ELOVL4 produce"s" was not observed.

Added accordingly.

12.PAGE 7

"only in the motor cortex but also in cingulate and piriform cortices, striatum, hippocampus, and lateral ventricle (LV) wall where periventricular glial cells are located (Supplementary Figure 4), similar to observations made in aging studies"

All these brain regions should also be used for determining the number of ELOVL4 expressing cells and reported.

We added the data in new Supplementary Figure 1.

13.PAGE 7 – 8

Discussion of ORO is incomplete regarding the what is seen in the figures 3 and Supplementary Fig 4.

Figure 3, ORO LD are difficult to resolve in current magnification of Fig. 3. No magnification insets are provided for Control tissue staining. Some lipid droplets may be visible in CA3 and striatum.

Supplementary Figure 4. Validation of the ORO staining using aging mouse brain ORO staining is difficult to resolve at currently presented magnification in supplemental Figure 4.

We included magnified images in Supplementary Figure 3 (original Supplementary Figure 4). In Figure 3, we provided those in control samples. To enhance visibility of LD, the squares that enclose focused areas were removed from images at lower magnification

14.Supplementary Figure 5. BODIPY signal should be quantified.

We added the quantification in Supplementary Figure 4 (original Supplementary Figure 5).

15.Supplementary Figure 6. Lipid droplets cannot be resolved at the magnification presented.

We now included higher magnification views in insets in new Supplementary Figure 5.

16.PAGE 9

"both motor learning deficit and anxiety phenotype that were demonstrated by a significant reduction in the learning index and time spent in the open arm, respectively (Figure 4b, c, d)." However, in the EPM test, it is not possible to determine if there is a motor deficit or anxiety

phenotype. If there is a motor deficit, an anxiety phenotype cannot be accurately determined. Because there is a decreased total number of arm entries, (open and closed), the motor deficit precludes any conclusions to be made about anxiety behaviors. Additional information about the motor deficit is required to make any conclusions about the anxiety behavior. Roto-rod is used to inform on the motor activities, not the learning behavior (PAGE 16). This indicates there is a motor deficit and the learning and anxiety behaviors cannot be determined.

As additional information for discussion of potential motor deficits, we cited our previous paper that examined the possibility of motor deficits using open field test. We also added detail of learning deficit but not motor coordination deficit observed in accelerated rotarod test at the end of the paragraph as follows: Line 211> In addition, the changes in these behaviors are less likely due to the locomotion issue, as the PAE model does not show any changes in the open field test compared to control mice in our previous study⁴⁴. Consistently, motor coordination evaluated at trial 1 does not show any differences between PAE and control either (Figure 4c)⁴⁴.

Reviewer #2 (Remarks to the Author):

In this manuscript, Hwang et al. show that prenatal exposure to the alcohol (PEA) will result in the postnatal changes in contributors to the fatty acid metabolism. For example, authors found upregulation of very long chain fatty acids protein 4 (Elovl4) and fatty acid 93 synthase (Fasn), and also of palmitic and arachidonic acid in the neocortical wall. Overall, authors elegantly show that PEA results in disrupted fatty acid metabolism in postnatal brain. Furthermore, authors report PEA induced accumulation of lipid droplets (LD) and changes in the gut microbiota and in studied motor and anxiety behaviors (e.g. accelerated rotarod test, and learning index). Overall, this is a strong study with an elegant experimental design that allowed accumulation of data and strong statistics that support main conclusions.

Thank you so much for reviewing manuscript. Our efforts were rewarded by receiving such uplifting remarks from the reviewer!

REVIEWERS' COMMENTS:

Reviewer #1 (Remarks to the Author):

This manuscript reports the findings that gene expression associated with fatty acid (FA) metabolism in the cerebral cortex of the PAE mouse model are altered. The revision includes important discussion about LD formation, as well as higher resolution of LD findings in mouse brain. They have also included appropriate discussion about motor function in the animal model in the context of the behavioral tests. They also now include relevant quantification of cell number. Minor revisions prior to publication are suggested below.

Minor suggestions:

Grammar concerns in lines 317 – 318 and 374-376.

381 indicates “suggesting no relation with neurobehavioral problems or lipid-associated brain pathology.” While line 385 states “of gut microbiota may profoundly and directly affect the brain pathology” This needs to be clarified.

Statistical analysis is appropriate.